



**The effect of water storage change in ET estimation in humid**
**catchments based on Budyko framework and water balance models**
Tingting Wang[1,2], Fubao Sun[1,2,3*], Hong Wang[1], Wenbin Liu[1], Hao Wang[4]
1. Key Laboratory of Water Cycle and Related Land Surface Processes, Institute of
Geographic Science and Natural Resources Research, Chinese Academy of Sciences,
Beijing, China
2. College of Resources and Environment, University of Chinese, Academy of
Sciences, Beijing, China
3. School of Civil Engineering, Hexi University, Zhangye City, Gansu Province,
China
4. China Institute of Water Resources and Hydropower Research, State Key
Laboratory of Simulation and Regulation of Water Cycle in River Basin, Beijing
100038, China
**Corresponding Author**: Fubao Sun (Sunfb@igsnrr.ac.cn), from Key Laboratory of
Water Cycle and Related Land Surface Processes, Institute of Geographic Sciences
and Natural Resources Research, Chinese Academy of Sciences

21 2017/03/27

**Keywords**: evapotranspiration, humid catchments, water storage change, water
balance, Budyko hypothesis





## Abstract

An accurate estimation of evapotranspiration (ET) in humid catchments is essential
in water-energy budget research and water resource management. While it remains a
huge challenge and there is no well accepted explanation for the difficulty of annual ET
estimation in humid region so far. Here we present the ET estimation in 102 humid
catchments over China based on the Budyko framework and two hydrological models:
abcd model and Xin'anjiang model, in comparison with ET calculated from the water
balance equation ($ET_{wb}$) on the ground that the $\Delta S$ is approximately zero at multiannual
and annual time scale. And we provide one possible explanation for the poorly annual
ET estimation in humid catchments as well. The results show that the Budyko
framework works fine in ET estimation in humid catchments at multi-annual timescale,
while neither the Budyko framework nor the hydrological models can estimate ET well
at annual timescale. One major cause for this poorly annual ET estimation is the
neglecting of $\Delta S$ in $ET_{wb}$ since it enlarges the variability of real actual ET. Much
improvement has been made when comparing estimated $ET + \Delta S$ with those $ET_{wb}$, and
the bigger the catchment area is, the better this improvement can be. This provides an
acceptable explanation for the poorly estimated annual ET and reveals the important
role of annual $\Delta S$ in ET estimation and validation in humid catchments. We highlight
that the annual $\Delta S$ shouldn't be taken as zero in water balance equation in humid
catchments.





## 1 Introduction


The evapotranspiration (ET) over terrestrial surface is the second largest
component of the global terrestrial water cycle since it returns about two thirds of
precipitation (P) that falls over the land. The partitioning of P into streamflow (Q) and
ET is captured by the ratio of potential evapotranspiration (PET) to precipitation ($\bar{\phi}$
=PET/P, dryness index), i.e., the Budyko hypothesis (Milly, 1994;Yang et al.,
2006;Donohue et al., 2007;Tekleab et al., 2011), and it is further constrained by physical
limits, namely, the limitation of available water (ET<P) in non-humid catchments and
limitation of available energy (ET<PET) in humid catchments (Fu, 1981;Milly and
Dunne, 2002). In non-humid catchments, the Budyko hypothesis and many other
approaches, e.g., abcd model (Sankarasubramanian and Vogel, 2002;Martinez and
Gupta, 2010) and remote sensing outputs (Zhang et al., 2008b), have been well
proceeded in ET estimation and thus Q prediction and water resource management
around the global (Ukkola and Prentice, 2013;Xu et al., 2013) and regional like
America (Chen et al., 2013), Australia (O'Grady et al., 2011) and China (Yang et al.,
2006;Yang et al., 2007). While in humid region, the quantity of research are limited
(Tekleab et al., 2011;Zhang et al., 2012;Carmona et al., 2016). Proportional relationship
is discovered between PET and ET (Wang and Hejazi, 2011;Cong et al., 2008) but the
estimated ET is less well based on the Budyko hypothesis and other models (Wang and
Hejazi, 2011;Zhu et al., 2013). Zhang et al (2012) compared ET based on the PML
model (Leuning et al., 2008;Zhang et al., 2012), products from Jung (Jung et al., 2010)





and the Budyko framework against ET from water balance equation ($ET_{wb}$) in 110
humid catchments around the global, but found that their $R^2$ were very small and the
trends didn't match well, showing that these approaches are, as yet, not sufficiently
accurate to explain ET in humid regions.
However, there is no widely well accepted explanations for the difficulty of annual
ET estimation in humid catchments so far as there are some unknown obstacles exist.
The poorly estimation of ET in humid catchments remains to be a huge challenge for
bridging the important gap in our knowledge of the hydrologic cycle. Hence,
researchers have made their attempts based on the commonly used Budyko framework
with one or more parameters included (Fu, 1981;Wang and Tang, 2014;Peter et al.,
2016;Carmona et al., 2016) to better estimate ET in humid catchments, and try to build
some kind of relationship with possible factors, e.g., vegetation (Donohue et al.,
2007;Ye et al., 2015), water storage change (Chen et al., 2013;Zeng and Cai, 2015a),
topographic factors (Xu et al., 2013) and integrative of various controls (Ye et al.,
2015;Chen et al., 2013) to further investigate the possible explanation.
The water storage change (ΔS), in particular, has been considered to be one
important factor that affects ET estimation in humid catchments. The soil water storage
performs as a filter from P to Q in catchments (Daly and Porporato, 2006), and this
filter effect and water storage change would introduce a bias into ET estimation,
especially in humid catchments. Milly (1994) discussed the effect of soil moisture on
ET theoretically and indicated a mediating effect on ET based on the Budyko



framework. Sound evidence suggests that the $\Delta S$ cannot be neglected at both annual
(Zhang et al., 2008a;Wang, 2012;Bai et al., 2016) and intra-annual (Chen et al., 2013;Ye
et al., 2015) timescales. Liu et al. (2016) specific present a worldwide evaluation of
nine ET products with consideration of $\Delta S$. Mao et al. (2016) found that the $\Delta S$ in
reservoirs altered the calculated ET trends of almost zero increase to a 4.2% increase
per decade when not taking $\Delta S$ into account over China. Moreover, the $\Delta S$ can affect
the variability of ET in humid catchment as well. (Wang and Alimohammadi,
2012;Chen et al., 2013;Ye et al., 2015;Zeng and Cai, 2015a) Zeng and Cai (2015a)
pointed out that the variability of the $\Delta S$ made some contribution to the variability of
intra-annual ET in humid catchments, through much smaller than that in non-humid
catchments (Wang et al., 2009;Wang and Zhou, 2015). Wang (2012) concluded that the
inter-annual variability of ET is not strongly correlated with P variability but with the
variations of P-$\Delta S$.
People try their best to improve the models with consideration of $\Delta S$ to meet $ET_{wb}$
so as to reveal the trend and pattern of ET. Specifically, improved version of the Budyko
framework of using P-$\Delta S$ as equivalent precipitation has been used and improvement
has been reported in ET estimation in non-humid catchments (Zhang et al., 2008a;Du
et al., 2016). However, there are not much reported in humid catchments based on this
kind of improvement. In addition, one would wonder why the results are always so poor
based on these approaches when compared with $ET_{wb}$.
From another angle, the $\Delta S$ is often taken as zero in $ET_{wb}$ on both multi-annual and





annual timescales (Donohue et al., 2007;Yang et al., 2007). In detail, the calculated
$ET_{wb}$ works both as real actual ET in hydroclimatology and as the standard actual
evapotranspiration to calibrate/validate ET based on other hydrological models. It
would prone to errors associated with ungauged subsurface runoff transfer and therefore
produce relatively unreliable estimation of ET (Zhang et al., 2008a;Wang, 2012). Hence,
unreliable input of $\Delta S$ would probability lead to an unexpected consequence to both
$ET_{wb}$ and modelled ET (Mao et al., 2016;Chen et al., 2013).

Here we are motivated to reveal one possible consequence caused by neglecting

$\Delta S$ in annual $ET_{wb}$ in humid catchments over China. The aim of this study is (1)
demonstrating the ET estimation based on the commonly used Budyko framework and
hydrological models, i.e., abcd model and Xin'anjiang model, when compared with
$ET_{wb}$ on the ground that the annual $\Delta S$ is approximately zero in 102 humid catchments
over China, and (2) presenting one possible explanation of poorly estimated annual ET
in humid catchments, and (3) revealing the importance of annual $\Delta S$ in $ET_{wb}$ calculation
and thus validation. Section 2 provides a brief summary of methods used in ET
estimation for humid catchments, while section 3 documents the data sources and
selected catchments used in this study. Results and discussion are presented in section
4, followed by the conclusion and summary in section 5.





## 2 Methodology

The water balance equation has been widely used in ET calculation and model
calibration/validation at multi-annual and annual timescales, since P and Q are
measured reliably in catchment hydrologic cycle and the ΔS is assumed to be zero
(Yang et al., 2006). For any time period, the water balance equation can be written as:
$$ET_{wb} = P - Q - \Delta S \tag{1}$$

Here, we first use $ET_{wb}$ as standard ET to validate modelled ET in humid catchments.

## 2.1 Hydrological models: abcd and Xin'anjiang models

However, the water balance equation is often limited within observed variables and
related hydrological models have been widely used. The abcd model is chosen for its
simplicity of monthly inputs and 4 parameters only, and Xin'anjiang model is for its
fine runoff simulation in humid catchments.
The abcd model is a well-known conceptual hydrological model with 4 parameters
(a, b, c, and d) developed by Thomas (1981) at annual timescale, and later tested and
recommended by Alley (1984) at monthly time step in water resources assessment
(Martinez and Gupta, 2010) and climate change (Sankarasubramanian and Vogel, 2002).
Compared with other sophisticated water balance models, the abcd model is quite
simple and the inputs are monthly P and PET while outputs include monthly Q, soil
moisture, groundwater storage and ET.





The partitioning of monthly $P_t$, which is determined by $PET_t$ and the initial storages
in soil moisture and groundwater, $S_{t-1}$ and $G_{t-1}$, into $Q_t$, $ET_t$, soil moisture storage $S_t$,
and groundwater storage $G_t$ is as follows:

$$Y_t(W_t) = \frac{W_t + b}{2a} - \sqrt{(\frac{W_t + b}{2a})^2 - \frac{W_t b}{a}}$$

(2)

where $Y_t$ is the sum of monthly evapotranspiration and soil moisture storage at the end
of the month, namely evapotranspiration opportunity, and $W_t$ is the sum of monthly
precipitation plus initial soil moisture, named as available water. The soil moisture at
the end of period t is written as:

$$S_t = Y_t \exp(-PET_t/b)$$

(3)

the ET at the period t is the difference between evapotranspiration opportunity and soil
moisture ($Y_t - S_t$). The $G_t$ and $Q_t$ are computed based on:

$$G_t = G_{t-1} + c(W_t - Y_t) - dG_t$$

(4)

$$Q_t = (1-c)(W_t - Y_t) + dG_t$$

(5)

the four parameters: a, the propensity for runoff to occur before the soil is saturated

to capacity, b, the upper bound of $Y_t$. c is equal to the fraction of streamflow, which
arises from groundwater, equivalent to the base flow index and d is proportional to the
base flow recession constant (Thomas, 1981). We adopt the abcd model to provide
monthly ET (hereafter $ET_{abcd}$) and $\Delta S$ (soil moisture change plus groundwater change)
in this study.

The Xin'anjiang model is a widely used lumped rainfall-runoff model developed

by Zhao et al. (1980) and Zhao (1992) consisting three sub-models, a three layer





evapotranspiration sub-model, a runoff generation sub-model and a runoff routing sub-
model (Zhao, 1992). The parameters and schematic diagram can be found in many
references (Li et al., 2009;Zhao, 1992) so we won't give any unnecessary details. It has
been widely used in runoff simulation and hydrological processes modelling in humid
and semi-humid regions (Rui et al., 2012). Here we use the Xin'anjiang model in ET
estimation in one selected typical humid catchment and in supporting the results from
abcd model.

## 175     2.2 The Budyko framework

The widely used Budyko framework, derived by Budyko(1963;1974) based on

findings of Schreiber (1904) and Ol'Dekop (1911), describes the water-energy balance
status of a catchment using the well-known "Budyko curve", which is empirically
derived based on energy supply (represented by PET) and water availability
(represented by P) on ET. Fu (1981) gave the differential forms and achieved the
analytical solutions of the Budyko hypothesis, providing a theoretical basis for the
Budyko framework. Subsequently analysis on annual water-energy balance have
proofed that the Fu's equation can be used in both long-term and annual water-energy
balances in non-humid catchments (Yang et al., 2007) and humid catchments as well
(Tekleab et al., 2011;Xu et al., 2013). The Fu's type of equation is as:

$$\frac{ET}{P} = f(\frac{PET}{P}) = 1 + \frac{PET}{P} - (1 + (\frac{PET}{P})^w)^{1/w}$$

(6)

where *w* is a dimensionless parameter related to the local factors. PET is estimated





based on the widely used FAO (Food and Agricultural Organization) Penman-Monteith

reference evaporation (FAO-Penman model) (Allen et al., 1998):

$$PET = \frac{0.408\Delta(R_n - G_s) + \gamma \frac{900}{T+273} u_2 e_s (1 - Rh/100)}{\Delta + \gamma(1 + 0.34u_2)} \tag{7}$$

where $R_n$ is net radiation (MJ/(m$^2$·day)), $\Delta$ is slope of the vapor pressure curve in kPa/℃,

$G_s$ is soil heat flux (MJ/(m$^2$·day)), $u_2$ (m/s) is the wind speed at 2 m height, $\gamma$ (kPa/℃)

is the psychometric constant, $e_s$ (kPa) is saturation vapor pressure at a given air

temperature, Rh is the relative humidity. We apply the standard algorithm to estimate

PET as per recommended by the FAO based on daily meteorological data series, namely,

wind speed, air temperature, sunshine duration and relative humidity.

A more accurate and quantitative understanding of ET variability is essential for

evaluation and validation of climate models and climate change impact assessment

(Zeng and Cai, 2015a ). Zeng and Cai (2015b) extended the theoretical framework,

which was first employed by Koster and Suarez (1999) to derived inter-annual ET

variability from variability of P, and accounted for the effect of the P, PET and ΔS on

intra- and inter-annual ET variability. A detailed annual ET variance based on the Fu's

type of Budyko framework is decomposed into variance/covariance of P and PET:

$$\sigma_{ET}^2 = w_p \sigma_P^2 + w_{PET} \sigma_{PET}^2 + w_{p,PET} \, \text{cov}(P, PET) \tag{8}$$

where σ and cov represent the standard deviation and covariance, and $w_p$, $w_{PET}$, and

$w_{p,PET}$ are the weighting factor quantifying the contribution from P, PET and their

interaction to ET variability with analytically calculation as:





$$w_p = (f(\bar{\phi}) - f'(\bar{\phi})\bar{\phi})^2$$
$$w_{PET} = (f'(\bar{\phi}))^2$$
$$w_{p,PET} = 2(f(\bar{\phi}) - f'(\bar{\phi})\bar{\phi})f'(\bar{\phi})$$

(9)


where $\bar{\phi}$ is the dryness index calculated based on the multiannual mean of P and
PET in each catchment.
We adopt the Budyko framework as one major approach to estimate ET and to
further analyze the ET variability so as to better understand ET in humid catchments.
**3 Data**
To demonstrate the ET estimation, existing problems and possible causes in humid
region, we choose 102 humid catchments ($\bar{\phi} < 1$) over southern part of China (Figure
1) with continuous monthly P, PET and observed Q (hereafter $Q_{obs}$) mainly for 1960-
2013 (some catchments miss several years of $Q_{obs}$ during this period while a few others
can trace back to 1950s): P, observed from daily meteorological dataset from China
Meteorological Data Network (http://data.cma.cn/); PET, calculated using equation 7
based on daily meteorological datasets containing the surface air temperature, sunshine
duration, wind speed and the relative humidity from China Meteorological Data
Network as well, and $Q_{obs}$, obtained from Annual Hydrological Report P. R. China:
hydrological data from Yangtze River Basin, Pearl River Basin and Southeast Rivers
Bain. The $\Delta S$, is obtained from the output of abcd model. Both P and PET are
interpolated and extracted at catchment scale and then form the monthly and annual
time series along with Q and $\Delta S$ in the following estimation and analysis.



What's more, the daily P, PET and $Q_{obs}$ for the selected typical catchment (black
bold line defined and railed out by hydrological station Dongbei in Figure 1) over 2001-
2012 are used for parameters calibration in Xin'anjiang model. And then we simulate
daily Q (hereafter $Q_{XAJ}$) and ET (hereafter $ET_{XAJ}$) for 1957-2013 (with available
observed monthly Q for this period as validation). The Xin'anjiang model is chosen for
its fine runoff simulation in humid catchment in order to further verify the results from
abcd model, which requires monthly inputs instead of that for the Xin'anjiang model at
daily timescale.

<Figure 1>

**4 Results and discussion**
We adopt the commonly used Budyko framework and hydrological models: abcd
model and Xin'anjiang model in ET estimation in humid catchments. First, we calibrate
the parameter *w* in Fu's type of Budyko framework (Figure 1) with annual P, PET, and
$Q_{obs}$ for each catchment, respectively. Then we estimate ET based on these approaches
and make comparison with $ET_{wb}$ on the ground that the ΔS is approximately zero at
both multiannual and annual timescales. The results of ET estimation in humid
catchments in 4.1 and 4.2 are at multiannual and annual time scales, respectively and
possible causes for poorly estimated annual ET is shown in 4.3.





## 4.1 ET estimation at multiannual timescale


There is common sense that the ΔS is approximately zero at long term water
balance (Donohue et al., 2007;Tekleab et al., 2011). We adopt the Fu's type of Budyko
formula and abcd model to estimate ET in 102 humid catchments over China and
compare with the $ET_{wb}$ at multiannual timescale. The relationship between the dryness
index ($\bar{\phi}$) and the evaporative index (ET/P) based on the Budyko framework for 102
selected humid catchments are plotted in Figure 2a, which fits the Budyko hypothesis,
showing a clear feature of energy control in these humid catchments.
The comparison of estimated ET using the Budyko hypothesis (denoted $ET_{budyko}$)
and abcd model against $ET_{wb}$ in humid catchments at multiannual timescale are shown
in Figure 2b. Excellent agreement has achieved with very high determination
coefficient ($R^2 \approx 1.0$) and slope of 0.95 when using the Budyko framework. However,
the comparison between $ET_{abcd}$ and $ET_{wb}$ shows that the abcd model works less well
with $R^2$ of 0.82 and the slope of 0.57, which underestimate the ET in humid region,
especially in those catchments where the ET are relatively large. As for those
catchments where ET are small, the abcd model works just as fine as the Budyko
framework. The Budyko framework is well recommended in ET estimation in humid
catchments at multiannual time scale since its reliability is greatest when applied using
long-term averages in both non-humid and humid catchments.

<Figure 2>






## 4.2 ET estimation at annual timescale

ET estimation at annual time scale has drawn more of our attention since it is

closely related to the runoff simulation, water resource management, etc. We adopt the

Fu's type of Budyko framework and hydrological models, i.e., the abcd model and

Xin'anjiang model in annual ET estimation and make comparison with $ET_{wb}$ on the

ground that the $\Delta S$ is zero at annual time scale, which is as common practice. The ET

estimation in humid catchment using the Budyko framework is in 4.2.1 and the results

based on the hydrological models mentioned above are in 4.2.2.

### 4.2.1 Annual ET estimation based on the Budyko framework

The Budyko framework works very well in ET estimation at multiannual time scale,

while whether it is the same or not at annual time scale remains unknown. Hence, we

adopt the Fu's type of Budyko framework first to demonstrate the estimation at annual

timescale. The parameter $w$ calibrated in 102 humid catchments are shown spatially in

Figure 1 and statistical in Figure 3a. The results show that the value of $w$ varies greatly

with maxima and minima of 4.16 and 1.33 and the majority distributed around 1.8~2.4

(25%~75% percentile), which indicating more about terrestrial feature (Figure 1). The

$R^2$ between $ET_{wb}$ and $ET_{budyko}$ (Figure 3b) in more than 90% of the humid catchments

over China are lower than 0.2 at annual time scale. Specifically, the average Root Mean

Square Error (RMSE) for 102 humid catchments is about 20.0% when comparing





$ET_{budyko}$ against their multiannual average of $ET_{wb}$. The ratio ($RMSE/ET_{wb}$) in about
half of the catchments are greater than 17% and more than 90% of humid catchments
are greater than 12.5% (Figure 3c). Hence, the Budyko framework works poorly in
humid catchments over China at annual timescale as some stressed (Zhang et al., 2012).

<Figure 3>

**4.2.2 Annual ET estimation based on the hydrological models**
The hydrological models provide as a useful tool in runoff simulation, ET, soil
moisture and groundwater storage estimation in both humid and non-humid catchments.
Since the Budyko framework works poorly in annual ET estimation in humid
catchments, we are attempt to present and evaluate how the commonly used
hydrological models work in ET estimation here at annual time scale.
First, we adopt the famous and widely used abcd model in estimating monthly and
thus annual ET in 102 humid catchments. The inputs includes the monthly P, PET and
the outputs are simulated monthly streamflow ($Q_{abcd}$), ET and $\Delta S$. The Nash-Sutcliffe
efficiency coefficient (NSE) between annual $Q_{abcd}$ and $Q_{obs}$ in these 102 selected humid
catchments are all higher than 0.65 (Figure 4), and mostly are around 0.8. However, the
$R^2$ between $ET_{abcd}$ and $ET_{wb}$ are all almost very small, i.e., smaller than 0.4 in each
corresponding catchments at annual time scale. And the $R^2$ in bout 60% of these humid





catchments are smaller than 0.1, showing poor ET estimation when using the abcd
model in humid catchments over China.

<Figure 4>

<Figure 5>


To further verify the results above, we adopt the well-known Xin'anjiang model,

which is famous for its well streamflow simulation in humid catchments, to estimate
ET in the selected catchment governed by hydrological station Dongbei (shown in
Figure 1). The inputs include continuously daily P and PET for 2001-2012, since we
have access to observed daily streamflow in this catchment for model calibration. The
NSE of daily $Q_{obs}$ and $Q_{XAJ}$ over this period is 0.786 in this humid catchment. We then
use the calibrated parameters to run Xin'anjiang model for 1957-2013 since monthly
$Q_{obs}$ is available for this period. The NSE of observed and simulated monthly
streamflow is 0.913 based on the Xin'anjiang model (Figure 5b), and 0.890 when using
the abcd model for 1957-2013 (Figure 5c). Strikingly, the NSE are 0.917 and 0.776
based on Xin'anjiang and abcd models in streamflow simulation at annual timescale.
Hence, both two models work fine in streamflow simulation in humid catchments and
the Xin'anjiang model achieve better. However, when it comes to annual ET estimation,
the $R^2$ are very small, 0.056 and 0.002 for $ET_{XAJ}$ and $ET_{abcd}$ against $ET_{wb}$ in this selected
catchment. The two hydrological models can well simulate streamflow while poorly in



annual ET estimation in this humid catchment. And we can easily speculate that the
Xin'anjiang model wouldn't work well in annual ET estimation in other humid
catchments as well. From above, the results based on the Xin'anjiang model well
support the estimation using the abcd model, and again verify the poorly ET estimation
in humid catchment at annual time scale.
In summary, both the Budyko framework and the hydrological models work poorly
in ET estimation in humid catchments when compared with $ET_{wb}$ on the ground that
the $\Delta S$ is approximate zero at annual time scale. These approaches are, as yet, not
sufficiently accurate in annual ET estimation in humid catchments. This poorly annual
ET estimation remains to be a universal challenge for bridging the important gap in our
knowledge of the hydrologic cycle.
**4.3 Potential reason for poorly-estimated annual ET in humid**
**catchments**
The above results at annual timescale is quite confusing. Since the P is observed,
the Q can be well simulated, and $\Delta S$ is zero as common practice, how come the annual
ET estimation is that terrible? From another perspective, there may some other not-
well-known problems exist in annual ET estimation in humid catchments more than
just the hydrological models. What if the $\Delta S$ shouldn't been seen as zero in water
balance at annual time scale? The unreliable input of $\Delta S$ would probability lead to an
unexpected    consequence    to    both    $ET_{wb}$    calculation    and    modelled    ET
calibration/validation. Since the majority of catchments in west are distributed in karst





region with ungauged subsurface runoff transfers (Figure 1), and the catchments in the
east and southeast along the coastal area, where urbanization expansion accompanied
with groundwater extraction, inter-basin water transfer, etc, which would definitely
introduce large bias to annual $\Delta S$. And the unreliable $ET_{wb}$ with assuming zero change
in annual $\Delta S$ may thus lead to irresponsible real actual evapotranspiration, and further
poorly modelled ET calibration and validation in humid catchments. Hence, a probable
explanation based on the role of the $\Delta S$ in annual $ET_{wb}$ is launched.
**4.3.1 Analytical explanation of the effect of neglecting $\Delta S$ to annual $ET_{wb}$**
At a given time of a catchment, we estimate the annual ET using annual P and PET,
based on the Budyko framework, and then make comparison against corresponding
$ET_{wb}$, i.e., P-Q, on the ground that the annual $\Delta S$ is zero (Figure 6). Extra deviation
would emerge when annual $\Delta S$ is not neglectable, and the comparison made above is
between $ET_{wb}$ and modelled ET+$\Delta S$. Moreover, the larger the $\Delta S$ is, the larger this extra
deviation could be. The annual $\Delta S$ is often seen as zero in non-humid catchments since
most of P turns into ET (Yang et al., 2007), leaving the majority of the rest P to Q and
very small proportion to $\Delta S$. Therefore it would be acceptable to take $\Delta S$ as zero in
$ET_{wb}$ calculation and calibration/validation in non-humid catchments. While it is
uncertain whether taking annual $\Delta S$ as zero is applicable in humid catchments. Besides,
the low $R^2$ between $ET_{budyko}$ and $ET_{wb}$ in humid catchments above can to some extent,
reflect the effect of annual $\Delta S$ to $ET_{wb}$.



<Figure 6>
<Figure 7>

To further explain the effect of ΔS to annual $ET_{wb}$ calculation and thus
calibration/validation, we first make analytical explanation as below. There is no
available observed annual ΔS at catchment scale, and inaccuracy and uncertainty exist
in almost all models to simulate ΔS so far. It would be acceptable to make simple
assumption that annual ΔS is linearly related to annual P as well, since clear linear
relation between annual P and observed annual Q (Figure S1). The $R^2$ in almost all the
humid catchments over China are higher than 0.6 and about 65% of the catchments are
higher than 0.75 (Figure S1). The relation between annual P and ΔS, Q are assumed as,

$Q = \alpha P$                    (10)

$\Delta S = \beta P$                    (11)

where β is a dimensionless parameter, and α is the runoff coefficient. The ratio of
estimated ET against $ET_{wb}$ is as,

$\frac{ET}{ET_{wb}} = \frac{ET}{ET + \Delta S} = \frac{1 - \alpha - \beta}{1 - \alpha}$                    (12)

which indicates the effect of ΔS to annual $ET_{wb}$ and thus modelled ET estimation and
calibration/validation. We further set the range of parameter β to be -0.5~0.5 and
runoff coefficient, α, to be 0~1.
Then we plot the change of this ratio in Figure 7a with both the change of parameter
β in the range of (-0.5, 0.5) and parameter α in the range of (0, 1). The range of this





ratio varies greatly, which indicates the effect of $\Delta S$ to $ET_{wb}$ variously. The smaller the
$\Delta S$ is, the more approaching to 1.0 this ratio could be, showing more insignificant effect
of $\Delta S$ to $ET_{wb}$ estimation and validation.

We choose a few typical value of parameter $\beta$, representing different amount of

$\Delta S$. The change of ratio in different partition of P into Q are shown in Figure 7b. Apart
from the various effect of $\Delta S$ to $ET_{wb}$, the runoff coefficient can affect the ratio as well.
We choose two runoff coefficient, 0.57 (average of runoff coefficients over 102 humid
catchments here, blue dash in Figure 7b) and 0.11 (average of runoff coefficients over
108 non-humid catchments in China, red dash, the relevant data are provided by Fubao
Sun (Yang et al., 2007)). The result shows that the larger the runoff coefficient is, the
greater effect of $\Delta S$ to $ET_{wb}$ under the same proportional of $\Delta S$ from P. Hence, it is
worth noticing that the neglecting of annual $\Delta S$ would add more inaccuracy and
uncertainty in $ET_{wb}$ and modelled ET estimation and calibration/validation in humid
catchments than that in non-humid catchments.
**4.3.2 The effect of $\Delta S$ in $ET_{wb}$ in humid catchments over China**

To verify the effect of annual $\Delta S$ to $ET_{wb}$, we adopt the estimated $ET_{budyko}$, the $\Delta S$

from abcd output and make the following comparison below. The $\Delta S$ here includes the
soil moisture change, groundwater change and other causes that lead to water loss other
than Q and ET. We first use the selected typical catchment mentioned above as case
study and then expand it to all the 102 humid catchments to test the effect of $\Delta S$ in
annual $ET_{wb}$.

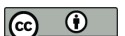



The time series of $ET_{wb}$, $ET_{budyko}$ and $ET_{budyko} +\Delta S$ for 1957-2013 in the selected

typical catchment (Figure 8) shows that through the $ET_{wb}$ and $ET_{budyko}$ are both

fluctuated around their multiannual average (about 761.8 mm/yr, 755.8 mm/yr), the

$ET_{wb}$ fluctuates more severe with variance of about 12518 $mm^2/yr^2$ than the variance

of $ET_{budyko}$, which is about 745 $mm^2/yr^2$. Interestingly, the comparison of the

variabilities of $ET_{wb}$ and $ET_{budyko}+\Delta S$ are quite similar, 12518 $mm^2/yr^2$ Versus 10611

$mm^2/yr^2$, and so does their multiannual average of about 761.8 mm/yr and 757.2 mm/yr

as well. Furthermore, the $R^2$ of original comparison between $ET_{wb}$ and $ET_{budyko}$ of 0.02

improves to 0.58 between $ET_{wb}$ and $ET_{budyko}+\Delta S$ when taking the $\Delta S$ into consideration.

This shows that the neglecting of $\Delta S$ can enlarge the fluctuations of annual $ET_{wb}$.

Therefore, the $\Delta S$ plays an important role in annual $ET_{wb}$ and thus in modelled ET

calibration and validation.

<Figure 8>

<Figure 9>

With the same approach, we expand this explanation to all 102 humid catchments

to verify the effect of neglecting annual $\Delta S$, and the results are shown in Figure 9. Much

improvement has been made when taking the $\Delta S$ into consideration in annual ET

validation instead of making direct comparison between $ET_{budyko}$ and $ET_{wb}$ in humid

catchments. For all 102 humid catchments categorized into three categories based on




their area, the original averaged $R^2$ between $ET_{wb}$ and $ET_{budyko}$ are smaller than 0.1 for
both small sized (41 catchments with the area all smaller than 5000 km$^2$) and moderate
sized catchments (33 catchments, and the area greater than 5000 km$^2$ but smaller than
10000 km$^2$), and about 0.12 for large sized catchments (28 catchments with area greater
than 10000 km$^2$). As for the $R^2$ within 10%~90% percentile, the small and moderate
sized catchments distribute around 0~0.15, and large catchments varies about 0~0.4.
The newly $R^2$ between $ET_{wb}$ and $ET_{budyko}+\Delta S$ promisingly show that, their average
improve to 0.18, 0.33 and 0.48 for three categories, respectively. In particular, the newly
$R^2$ within 10%~90% percentile, all showing satisfactory improvements with about
0.02~0.38 for small sized catchments, 0.11~0.50 for moderate sized catchments and
0.25~0.65 for large sized catchments. All above indicates that when using annual $ET_{wb}$
as standard ET and thus assessing modelled ET in humid catchments, the $\Delta S$ should not
be seen as zero in water balance equation. Besides, the larger the catchment area is, the
better estimation of $ET+\Delta S$ compared with $ET_{wb}$ in annual ET assessment.

<Figure 10>

For further interpretation, we present the variabilities of P, PET, $ET_{wb}$, $ET_{budyko}$ and
$ET_{Budyko}+\Delta S$ using the results of 102 humid catchments (Figure 10a), and variance
decomposition results based on Equation 9 (Figure 10b). The variability of P varies
mainly about 20,845~97,071 mm$^2$/yr$^2$ within 10%~90% percentile for 102 humid




catchments. While the range is only about 1,465~4,008 $mm^2/yr^2$ for the variability of
PET, which shall limit the variability of ET in humid catchments since it is controlled
by PET (Fu, 1981). The variability of $ET_{budyko}$ meets this limitation with min-max of
186~2,414 $mm^2/yr^2$, and 412~1355 $mm^2/yr^2$ within 10%~90% percentile for 122 humid
catchments. While the variability of $ET_{wb}$ (min-max of 2,835~50,114 $mm^2/yr^2$, and
7,161~26,142 $mm^2/yr^2$ within 10%~90% percentile) goes far beyond this limitation of
PET variability (Figure 10a). This may because most of the P variability transfers to the
variability of Q, and the majority of the rest transfers to the variability of ΔS and
covariance between relative components. This is in line with Wang and Alimohammadi
(2012) and very small proportional of P variability is left to the variability of ET, which
is further controlled by the variability of PET. This proves that PET not only controls
how much the ET is in a humid catchment, but also its variability. The annual $ET_{wb}$
with assuming zero change in ΔS would in fact enlarge the variability of real ET greatly
in humid catchments.
Furthermore, the comparisons of variance of $ET_{wb}$, $ET_{budyko}$ (Var(ET) in Figure 10b)
against variance of sum of three components: the decomposed variance of P, PET and
their covariance (Var($ET_{cal}$) in Figure 10b) further prove this conclusion. The Var($ET_{cal}$)
is almost linearly correlated with variance of $ET_{budyko}$ with $R^2$ of 0.966 and slope of
almost 1.0 in 102 humid catchments. However, the $R^2$ and the slope become 0.005 and
1.61 when compared Var($ET_{cal}$) with the variance of $ET_{wb}$, showing that the neglecting
annual ΔS changes the variability of $ET_{wb}$ in almost all the humid catchments. In



summary, the $ET_{wb}$ with assuming that the annual $\Delta S$ is zero, is not the accurate actual
evapotranspiration in humid catchments, and thus not suitable for modelled ET
assessment.

## 5 Conclusion and summary


Attempts on improving the ET estimation in humid catchments based on the
Budyko framework or other hydrological models have been made to meet the $ET_{wb}$ on
the ground that $\Delta S$ is zero at multiannual and annual timescales. While not much
improvement has been achieved, and an accurate estimation of annual ET in humid
catchments remains to be a huge challenge. In this research, we adopt the commonly
used Budyko framework and hydrological models, i.e., the abcd model and Xin'anjiang
model in ET estimation in 102 humid catchments over China. We are motivated to
explore the possible explanation of poorly annual ET estimation in humid catchments
from another perspective: the consequence caused by neglecting annual $\Delta S$ in $ET_{wb}$.
We present the estimated ET in humid catchments and make comparison with $ET_{wb}$
on the ground that the multiannual and annual $\Delta S$ are approximate zero, the same as
common practice. At multiannual timescale, the Budyko framework works very well
while the abcd model underestimate the ET to some extent, especially in those
catchments where ET are large (Figure 2). Hence, the Budyko framework is well
recommended in ET estimation in humid catchments at multiannual timescale. While
at annual time scale, both the Budyko framework and two hydrological models work



poorly, with very small $R^2$ in all cases when compared with $ET_{wb}$ (Figure 3~5), showing
that these approaches are not sufficiently accurate in annual ET estimation in humid
catchments. From another perspective, there may some other not-well-known problems
exist in annual ET estimation in humid catchments more than the models.
To explore the possible explanation for the poorly estimate annual ET in humid
catchments, we set out from the effect of neglecting annual $\Delta S$ in water balance
equation and adopt ET from the Budyko framework ($ET_{budyko}$) and $\Delta S$ from abcd output.
We make comparison of $ET_{budyko}$ and $ET_{budyko}+\Delta S$ against $ET_{wb}$ and find that much
improvement has been made when comparing $ET_{budyko}+\Delta S$ with $ET_{wb}$ (Figure 8~9). The
larger the catchment area is, the better this improvement could be. The $ET_{wb}$ enlarges
the variability of real actual evapotranspiration due to the variability of the $\Delta S$ (Figure
10), since PET controls both the ET and its variability in humid catchments. Hence, the
neglecting of annual $\Delta S$ in $ET_{wb}$ leads to the unreliable real ET in humid catchments,
and this is one major reason for the poorly estimated results since they are all validated
against $ET_{wb}$.
Above all, we are highlighting that the annual $\Delta S$ shouldn't be seen as zero in water
balance equation in humid catchments. The $ET_{wb}$ with original assumption that annual
$\Delta S$ is approximate zero is not the accurate ET and thus not suitable for modelled ET
calibration and validation in humid catchments.

**Acknowledgements**





This research was supported by the National Key Research and Development Program
of China (2016YFA0602402 and 2016YFC0401401), the CAS Pioneer Hundred
Talents Program (Fubao Sun), an Open Research Fund of State Key Laboratory of
Desert and Oasis Ecology in Xinjiang Institute of Ecology and Geography, Chinese
Academy of Sciences (CAS), and the International Science and Technology
Cooperation Program of China (2014DFA71910), the CPSF-CAS Joint Foundation for
Excellent Postdoctoral Fellows and National Science Foundation of China (41601035
and 41401037)

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



**Figure captions**
Figure 1 Spatial distribution of humid catchments ($\bar{\phi}$ > 1) over southern part of China
along with their corresponding parameter $w$ in Fu' equation, and one selected typical
catchment used as case study accompanied with its controlling hydrological station:
Dongbei Station.

Figure 2 The Budyko framework for 102 humid catchments over China in (a), and the
comparison of $ET_{wb}$ (ET calculated based on water balance equation) against $ET_{budyko}$
(ET estimated based on Fu's equation) and $ET_{abcd}$ (ET estimated using abcd model) at
multi-annual timescale for these humid catchments in (b).

Figure 3 The statistical information for estimated $ET_{Budyko}$ in 102 humid catchments: (a)
the calibrated parameter $w$ in the Fu's equation, (b) the $R^2$ between annual $ET_{wb}$ and
$ET_{budyko}$, and (c) the ratio of their RMSE and $ET_{wb}$ (RMSE/$ET_{wb}$) in percentage.

Figure 4 the Nash-Sutcliffe efficiency coefficient (NSE) between observed and
simulated (based on the abcd model) annual Q along with corresponding $R^2$ of annual
$ET_{wb}$ and $ET_{abcd}$ in 102 humid catchments over China.

Figure 5 The time series of monthly P over 1957-2013 for the selected typical catchment
in (a), and (b) the corresponding observed monthly streamflow ($Q_{obs}$) and simulated





one based on the Xin'anjiang model ($Q_{XAJ}$), along with the observed and modelled Q
using the abcd model ($Q_{abcd}$) in (c).

Figure 6 The schematic of $\Delta S$ in Budyko framework and water balance equation in
humid catchments (energy limited). ET is estimated based on given P and PET, and
validated against $ET_{wb}$ based on P and $Q_{obs}$ on the ground that the $\Delta S$ is zero, which can
affect the ET validation to some extent.

Figure 7 The effect of $\Delta S$ to $ET_{wb}$ on various proportion of Q and $\Delta S$. (a) The changes
of ratio ($ET/(ET+\Delta S)$) with the change of parameter p1 in range of (-0.5, 0.5) in vertical
axis, and parameter p2 of (0, 1) in horizontal axis (parameters p1 and p2 are in equations
10 and 11, respectively). The value of ratio ($ET/(ET+\Delta S)$) in range of (0, 2) are colored
as color bar, and value greater than 2 in right-bottom triangle area is set as wine red. (b)
The changes of ratio with the change of runoff coefficient (i.e., parameter p2) in several
selected typical proportion of $\Delta S$ (i.e., parameter p1), the red dash is the chosen runoff
coefficient representing the effect of $\Delta S$ to ET in non-humid ($\bar{\phi}$ >1) region, and blue
dash for humid ($\bar{\phi}$ <1) region.

Figure 8 The annual time series of $ET_{wb}$, $ET_{budyko}$ and $ET_{Budyko} + \Delta S$ over 1957-2013
for the selected typical catchment.

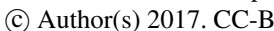


Figure 9 The box plot of comparison of $R^2$ between $ET_{budyko}$, $ET_{budyko}+\Delta S$ against
$ET_{wb}$, categorized by catchment area.

Figure 10 The statistics of variability of annual P, PET, $ET_{wb}$, $ET_{budyko}$ and $ET_{budyko}+\Delta S$
for 102 humid catchments in (a), and (b) the comparison of calculated variance based
on variance decomposition equation ($Var(ET_{cal})$) against the variance calculated from
$ET_{wb}$ and $ET_{budyko}$, respectively.

**Figure**

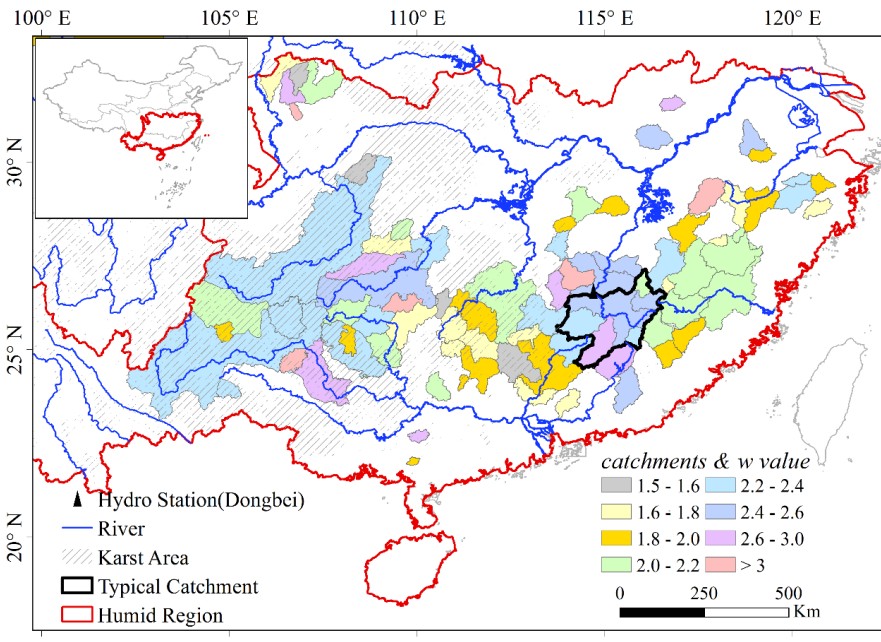


Figure 1 Spatial distribution of humid catchments ($\bar{\phi} > 1$) over southern part of China
along with their corresponding parameter $w$ in Fu' equation, and one selected typical




catchment used as case study accompanied with its controlling hydrological station:
Dongbei Station.

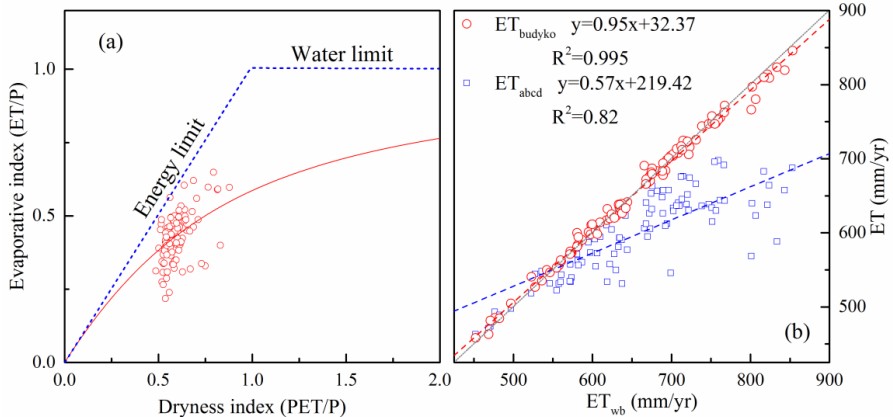


Figure 2 The Budyko framework for 102 humid catchments over China in (a), and the
comparison of $ET_{wb}$ (ET calculated based on water balance equation) against $ET_{budyko}$
(ET estimated based on Fu's equation) and $ET_{abcd}$ (ET estimated using abcd model) at
multi-annual timescale for these humid catchments in (b).

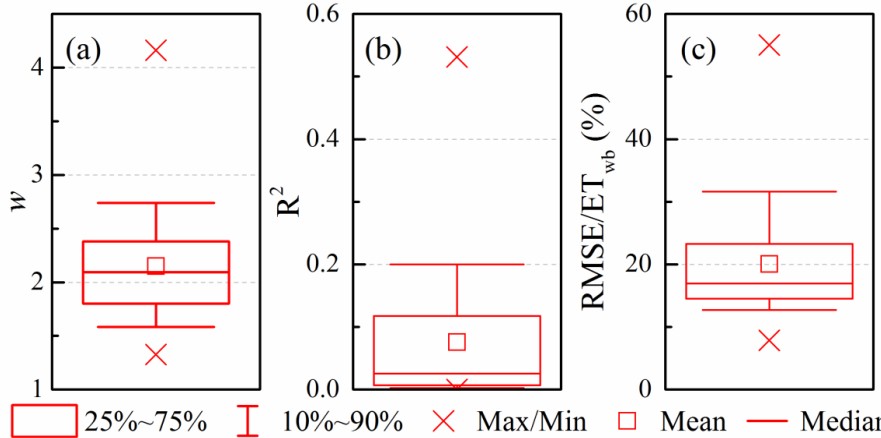






Figure 3 The statistical information for estimated $ET_{Budyko}$ in 102 humid catchments: (a)
the calibrated parameter $w$ in the Fu's equation, (b) the $R^2$ between annual $ET_{wb}$ and
$ET_{budyko}$, and (c) the ratio of their RMSE and $ET_{wb}$ (RMSE/$ET_{wb}$) in percentage.

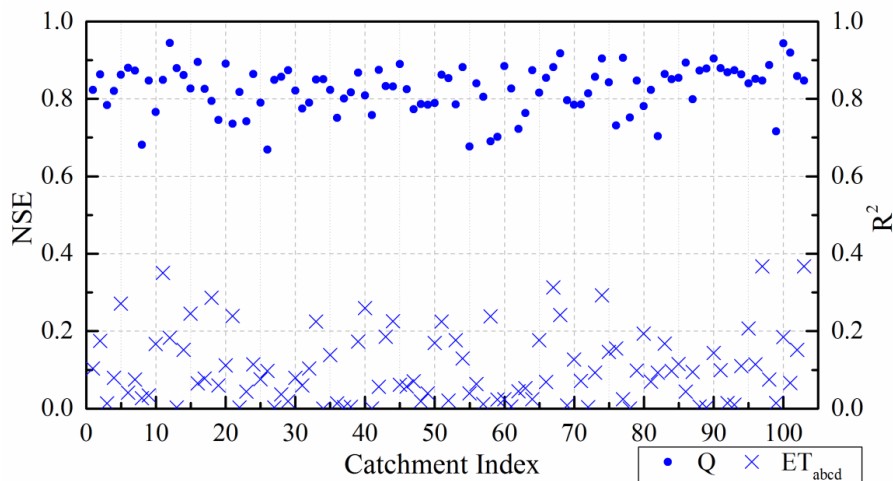


Figure 4 the Nash-Sutcliffe efficiency coefficient (NSE) between observed and
simulated (based on the abcd model) annual Q along with corresponding $R^2$ of annual
$ET_{wb}$ and $ET_{abcd}$ in 102 humid catchments over China.




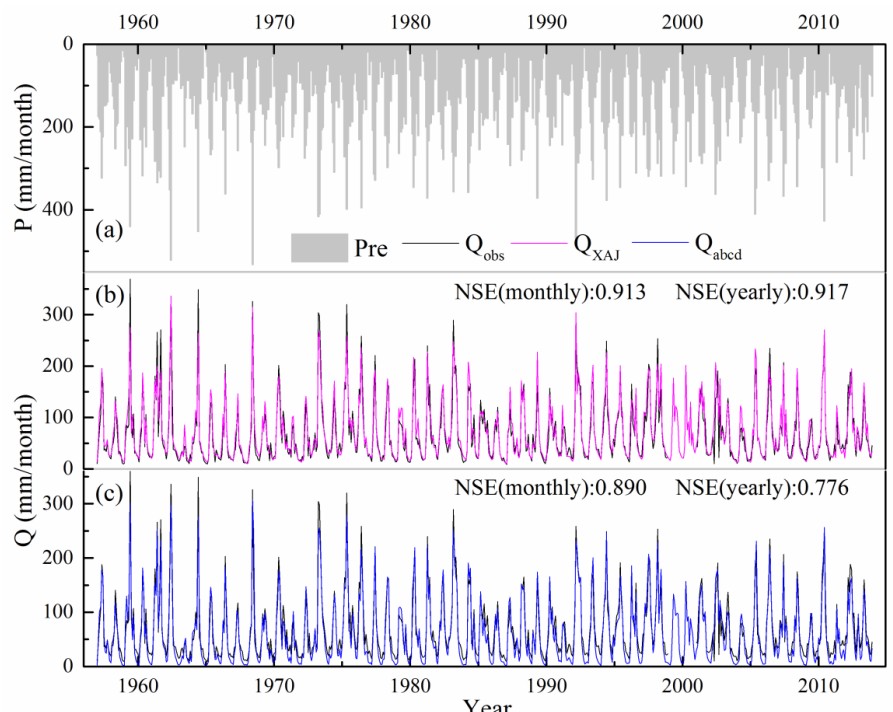


Figure 5 The time series of monthly P over 1957-2013 for the selected typical catchment
in (a), and (b) the corresponding observed monthly streamflow ($Q_{obs}$) and simulated
one based on the Xin'anjiang model ($Q_{XAJ}$), along with the observed and modelled Q
using the abcd model ($Q_{abcd}$) in (c).





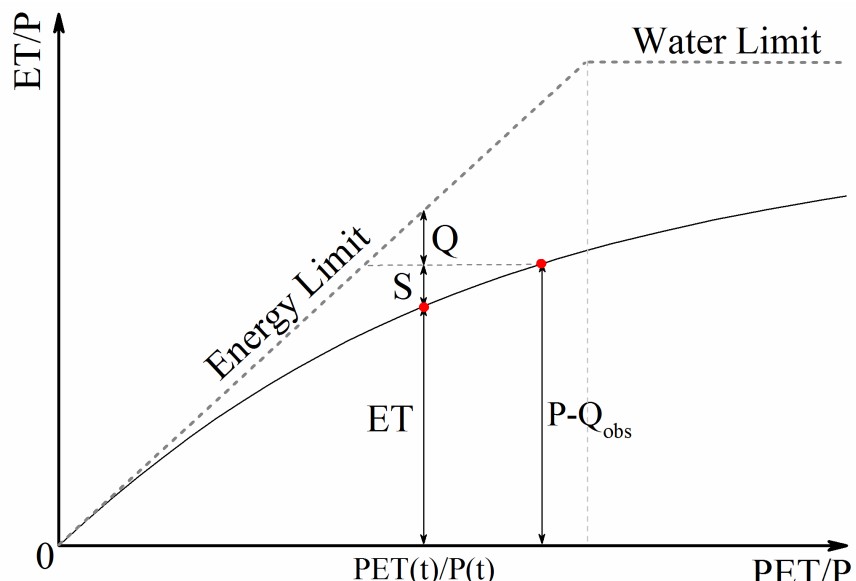


Figure 6 The schematic of ΔS in Budyko framework and water balance equation in
humid catchments (energy limited). ET is estimated based on given P and PET, and
validated against $ET_{wb}$ based on P and $Q_{obs}$ on the ground that the ΔS is zero, which can
affect the ET validation to some extent.

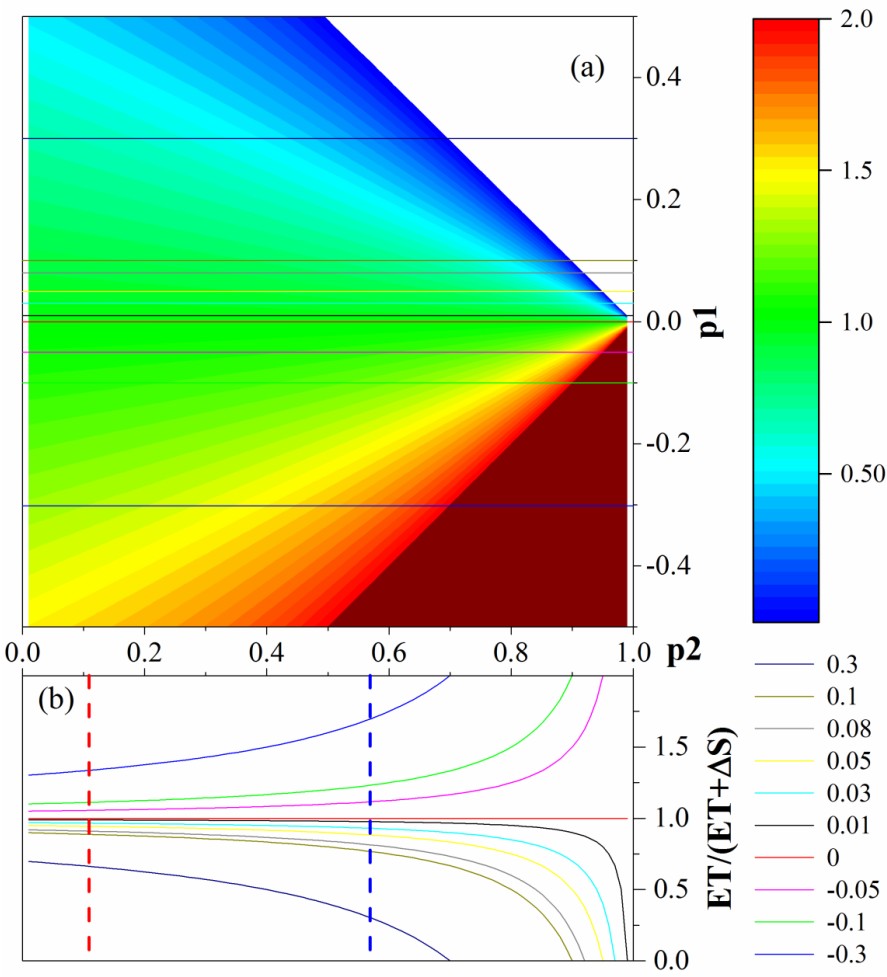


Figure 7 The effect of ΔS to ET$_{wb}$ on various proportion of Q and ΔS. (a) The changes
of ratio (ET/(ET+ΔS)) with the change of parameter p1 in range of (-0.5, 0.5) in vertical
axis, and parameter p2 of (0, 1) in horizontal axis (parameters p1 and p2 are in equations
10 and 11, respectively). The value of ratio (ET/(ET+ΔS)) in range of (0, 2) are colored
as color bar, and value greater than 2 in right-bottom triangle area is set as wine red. (b)
The changes of ratio with the change of runoff coefficient (i.e., parameter p2) in several
selected typical proportion of ΔS (i.e., parameter p1), the red dash is the chosen runoff




coefficient representing the effect of $\Delta S$ to ET in non-humid ($\bar{\phi} > 1$) region, and blue
dash for humid ($\bar{\phi} < 1$) region.

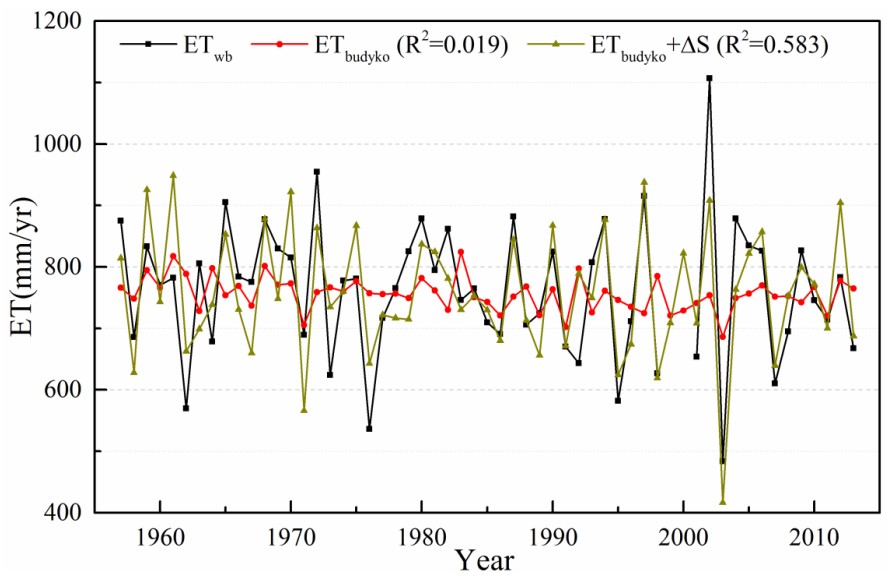


Figure 8 The annual time series of $ET_{wb}$, $ET_{budyko}$ and $ET_{Budyko} + \Delta S$ over 1957-2013
for the selected typical catchment.




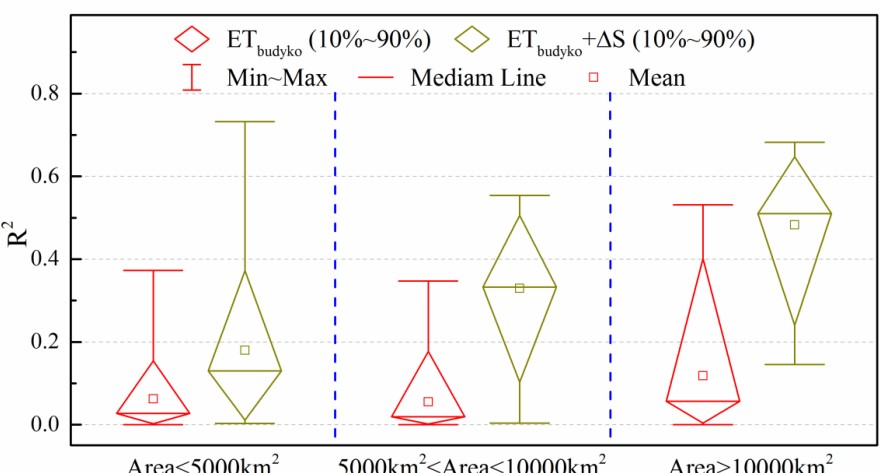


Figure 9 The box plot of comparison of $R^2$ between $ET_{budyko}$, $ET_{budyko}+\Delta S$ against
$ET_{wb}$, categorized by catchment area.

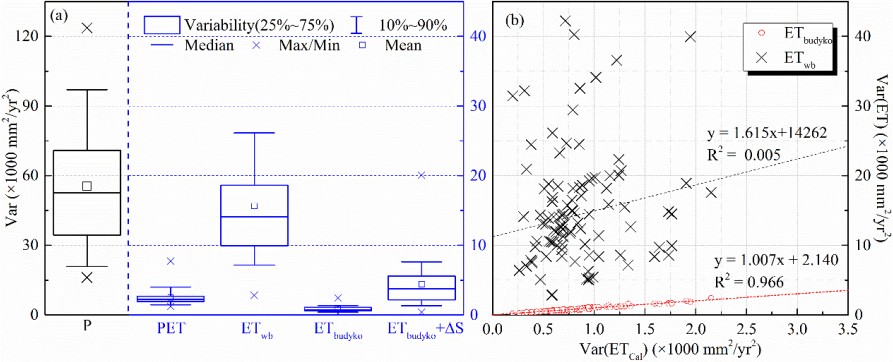


Figure 10 The statistics of variability of annual P, PET, $ET_{wb}$, $ET_{budyko}$ and $ET_{budyko}+\Delta S$
for 102 humid catchments in (a), and (b) the comparison of calculated variance based
on variance decomposition equation (Var($ET_{cal}$)) against the variance calculated from
$ET_{wb}$ and $ET_{budyko}$, respectively.



## 787 **List of notations**

| Variable name | Variable description | Units |
| --- | --- | --- |
| water balance components | | |
| P | Precipitation | mm |
| PET | Potential evaporation | mm |
| ET | Evapotranspiration | mm |
| Q | Streamflow | mm |
| $\Delta S$ | Water storage change | mm |
| $\bar{\phi}$ | Dryness index | |
| $\alpha$ | Runoff coefficient | |
| $\beta$ | Proportional of $\Delta S$ to P | |
| *w* | parameter in Fu's equation | |
| abcd model | | |
| a | Propensity for runoff to occur before the soil is saturated to capacity | |
| b | Upper bound of $Y_t$ | |
| c | Base flow index | |
| d | Proportional to the base flow recession constant | |
| $S_t$ | Soil moisture storage at the end of period t | mm |
| $G_t$ | Groundwater storage at the end of period t | mm |
| $Y_t$ | Evapotranspiration opportunity at the end of period t | mm |
| $W_t$ | Available water at the end of period t | mm |
| FAO-Penman model | | |
| $R_n$ | Net radiation | $MJ/(m^2 \cdot day)$ |
| $G_s$ | Soil heat flux | $MJ/(m^2 \cdot day)$ |





| $\Delta$ | Slope of the vapor pressure curve | kPa/℃ |
|---|---|---|
| $u_2$ | Wind speed at 2 m height | m/s |
| $\gamma$ | Psychometric constant | kPa/℃ |
| $e_s$ | Saturation vapor pressure | kPa |
| Rh | Relative humidity | |
| $w_p$ | Weighting factor for contribution of P to ET variability | |
| $w_{PET}$ | Weighting factor for contribution of PET to ET variability | |
| $w_{p,PET}$ | Weighting factor for contribution of interaction between P and PET to ET variability | |
