# Peer review of "The effect of water storage change in ET estimation in humid catchments based on Budyko framework and water balance models"

_Hydrology and Earth System Sciences, 2017_

## Referee Comment (RC1) · Anonymous Referee #1 · 21 Apr 2017

Review of "The effect of water storage change in ET estimation in humid catchments based on Budyko framework and water balance models" by Wang et al – submitted to HESS.

Wang and colleagues compare various ET estimation at the annual and multi-annual timescale and conclude that storage changes should be accounted for in ET estimations.

The writing of this paper needs a lot of improvement, and can benefit strongly from a native speaker rigorously rewriting it. I made a long list of suggestions for the first 50 sentences, but this list is not exhaustive (even not for this small part of the manuscript). In its current stage I cannot fully judge the scientific merit of the contribution because

the I do not always understand the interpretation of generated results.

The paper does not address anything sufficiently novel. The main conclusion that water storage changes can be significant at annual time-scales is already well known. In addition, some of the methods that the paper uses to estimate ET (Budyko and a water balance accounting) have been extensively reported to be only applicable at longer time-scales. Consequently, I do not see why the paper is relevant to HESS

The paper does not review previous work appropriately. Statements like "While in humid region, the quantity of research are limited (Tekleab et al., 2011;Zhang et al., 2012; Carmona et al., 2016)" do not represent the vast work that has been done in ET research. This also related to the previously addressed issue that the novelty of this paper is not sufficient.

The manuscript is not based on sound reasoning that can be easily followed. While this may be a linguistic issue, since the authors are (I assume) no native English speakers, it hinders me from properly judging the scientific merit of this work.

Sorry I cannot be more positive about the paper

Detailed comments - Title: Avoid abbreviations in the title, and write evapotranspiration instead of ET - Title: the title should be include the time-scale(s) that the paper reports on (e.g. annual) - Title: "for evapotranspiration estimations" instead of "in ET estimation" - Title: "water balance models" is unclear. I suggest to use more commonly used wording. - Title: "based on the Budyko framework" instead of "based on Budyko framework" - Line 26: "water-energy budget research" is unclear. - Line 26: "resources" instead of "resource" - Line 26-27: "while it", be explicit where "it" refers to. - Line 26 – 28: this sentence needs to be reformulated. - Line 28: include the timescale of ET estimates. - Line 29: "located in" instead of "over" - Line 28 – 32: "Here we ... time scale". This sentence should be completely rewritten. Consider to break down this information into two separate (shorter) sentences. - Line 32: "and we" is not a way to start a sentence. - Line 34: "works fine" is vague - Line 34: replace "in" by "for" - Line 35: be more

specific than "well" - Line 36: "of this poor" instead of "for this poorly" - Line 37: "the neglecting of dS" should be replaced by something like "the assumption that storage changes are negligible" - Line 37: "real" is redundant - Line 37: "increases" instead of "enlarges" - Line 37: specify if "it" refers to "neglecting storage changes" or "storage changes" - Line 38: "Much improvement has been made" - Line 37-39: This sentence should be rigorously rewritten: "Much improvement has been made when comparing estimated ET + $\Delta$S with those ETwb, and the bigger the catchment area is, the better this improvement can be." The message is clear, but it the sentence is not English. - Line 39-40: what does "an acceptable explanation" mean? Because of the confused English I am unsure how "certain" this attribution is. - Line 41-43: "We highlight ... humid catchments." Or "Overall this highlights that storage changes affect evapotranspiration at the annual time-scale and should therefore be accounted for." - Line 46: "over terrestrial surface" is redundant since terrestrial and land are repeated later in the sentence. - Line 47: replace "returns" for "equals" - Line 49: "is captured by" is vague. Do you mean mean "controlled by" - Line 50: replace "i.e.," by " as described by" - Line 50: Why is there no citation to the original Budyko paper/book? - Line 51: ", and it" it is unclear what "it" refers to + you should start this as a new separate sentence. - Line 52: the limitation you refer to are explained in the original Budyko equation, there is no need to refer to other studies here as long as this original publication is cited.

---

## Referee Comment (RC2) · F. Jaramillo (Referee) · 2 May 2017

The authors perform a multi-method assessment of ET in wet basins in Southern China and attempt to assess the role of assuming change in water storage as negligible (DS=0) in their calculations at the annual and muti-annual time scales. They also propose that this assumption may be the cause of poor annual ET estimates from three models (Fu, abcd and Xin'anjiang model) when compared to the ET from the water budget estimate at the annual scale. They find that including the estimates of Delta S at the annual scale from the abcd model reduces the variability of ET predictions. I think the subject is interesting and is nowadays gaining a lot of attention due to the wide use of the Budyko-type studies. The use of the abcd model to calculate DS sounds like

a good idea, and to compare ET Budyko with ET Budyko + △S. The scientific basins of this study is sound, at least from what I managed to extract from the extremely-difficult-to-understand manuscript. The exploration of change in water storage is interesting and resourceful (Fig. 7-10).

But I have some concerns that must be fixed in this manuscript:

1. Language. As it is now, I think the manuscript is unpublishable in HESS or any other decent scientific journal. The use of English is of poor quality. Some parts of the manuscript cannot be judged with scientific criteria because it is just impossible to understand what the authors are referring to. I myself do not have English as a native language, so I understand how frustrating it is to express your findings in another language. However, this manuscript needs to be written from beginning to end with the aid of a native English speaker. I recommend taking it to a professional writer or similar. I started fixing grammatical issues, but then I realized there was no case in doing this and I should better focus on the science.

2. Literature review on water storage and Budyko needs to be improved and updated. The authors have omitted important pieces of research dealing with the importance of water storage changes within the Budyko framework and specifically in "wet" regions. Some examples,

-[Moussa and Lhomme, 2016] – This study should give some insight on the possible mathematical formulations that the authors could further explore, apart from their multi-method assessment. -[Jaramillo and Destouni, 2015] – I know that the basins that the authors are studying are "humid", so irrigation is highly unlikely. However, flow regulation by water impoundment in reservoirs or water transfers affect the evaporative ratio ET/P in the long term, and it is not due to the water stored in the reservoirs as the authors suggest by citing Mao (2016). They find that flow regulation acts like a proxy of land and water use that can explain ET/P changes more than PET/P changes. -[Destouni et al., 2013] – Budyko analysis cannot get "wetter" than in cold and wet

[Figure]

Sweden. This study shows that accounting for surface water storage changes at the annual scale does not change long-term trends in basin-scale ET/P. -[Gudmundsson et al., 2016] – They find that changes in water availability are only dominated by changes in the aridity index in very humid climates.

3. I think the authors need to be clearer in what they refer with DeltaS. Moisture S and storage S. How can we know which is which. Is $\Delta$S=$\Delta$S+$\Delta$G.

4. I think they should justify their work in a better way. Saying that the effect of storage change in ET calculations in wet regions is scarce is just not true.

5. What is the multiannual scale? How did you measure DS (storage, not moisture) at this scale? How is it different from the annual scale? This is not clear at all. You never specify the periods

6. Line 420 to 421. The improvement of R2 from 0.02 to 0.58 sounds quite extraordinary. You should show this scatter plot due to its importance.

Other issues:

49. controlled instead of captured 53. What is humid= PET/P<1? 58. around the globe. . . and scales like 60. region(s), research is limited 61-64 Could not understand this 64-66 This is because these models do not account for all drivers of change in ET/P. For instance, look at [Jaramillo and Destouni, 2014]. 74% of all movements in Budyko space cannot be explained by only changes in PET/P during long periods of time.

Jaramillo, F., and G. Destouni (2014), Developing water change spectra and distinguishing change drivers worldwide, Geophys. Res. Lett., 41(23), 8377–8386, doi:10.1002/2014GL061848.

L. 71 Delete "exist" 81-82 This is obvious. 86 theoretical ET 89 specifically 90. The Mao study has one big problem. They calculate $\Delta$"ET"= $\Delta$P-$\Delta$R-$\Delta$S and not $\Delta$"ET"=$\Delta$P-$\Delta$R-$\Delta$($\Delta$S), which is incorrect. That is why they get such a big effect of storage change

in the ET calculations. 105. Again, this is because not all drivers of change are taken into account with Budyko-type models. 111 "It would prone"? 183 proved 224 Is DS = DG + DS? You have to differentiate storage from soil moisture, they are both S and confuses 257. Excellent? Change this word 282 Terrestrial feature? 303. Wy do you use the NSE. For what? Explain 306. You should state p-values for all R2 values that you give 342-349 Improve language. Impossible to understand due to the language. 356 "launched" Figure 6- Something does not fit here. ET+S+Q=ET/P?? Sometimes I feel that there is a confusion between variability, variance and R2. Please check this along the manuscript.

References

-Destouni, G., F. Jaramillo, and C. Prieto (2013), Hydroclimatic shifts driven by human water use for food and energy production, Nat. Clim. Change, 3(3), 213–217, doi:10.1038/nclimate1719.

-Gudmundsson, L., P. Greve, and S. I. Seneviratne (2016), The sensitivity of water availability to changes in the aridity index and other factors—A probabilistic analysis in the Budyko space, Geophys. Res. Lett., 2016GL069763, doi:10.1002/2016GL069763.

-Jaramillo, F., and G. Destouni (2015), Local flow regulation and irrigation raise global human water consumption and footprint, Science, 350(6265), 1248–1251, doi:10.1126/science.aad1010.

-Moussa, R., and J.-P. Lhomme (2016), The Budyko functions under non-steady-state conditions, Hydrol Earth Syst Sci, 20(12), 4867–4879, doi:10.5194/hess-20-4867-2016.
* * *

---

## Referee Comment (RC3) · Anonymous Referee #3 · 9 May 2017

**General comments**

This study analyzes the effect of changes in water storage at annual and multi-annual timescales for humid catchments in China. The authors perform a quantitative comparison of evapotranspiration (ET) estimates from the water balance, Budyko's framework, and two other models. They show that it is erroneous to obtain annual ET estimates from the water balance when neglecting changes in storage. I find the approach interesting, but numerous points need to be addressed. Particularly, the quality of the text needs to be greatly improved before being able to thoroughly assess the scientific merit of the study.

**Specific comments**

It is not right to assume the validity of Budyko for annual timescales. The sentence: "Subsequently analysis on annual water-energy balance have proofed that the Fu's equation can be used in both long-term and annual water-energy balances in non-humid catchments (Yang et al., 2007) and humid catchments as well (Tekleab et al., 2011; Xu et al., 2013)", is not accurate. Both Tekleab et al. (2011) and Xu et al. (2013), together with many other studies (e.g. Gentine et al. 2012; Roderick and Farquhar 2011) indicate that the validity of Budyko's framework requires steady-state conditions, which are generally achieved by using data at time scales significantly longer than 1 year.

I think it would be better if the storyline focuses from the beginning on the issue of neglecting ΔS for ETwb.

The approach for analyzing the inter-annual variability of ET needs to be clearer (equations 8 and 9). Is the effect of ΔS not accounted for? I believe it would be better to not include this section in the paper, and consequently Figure 10b. The authors already convey your point about the higher variability of ETwb compared to the other estimates of ET in Figure 10a. They could strengthen this argument by showing the histograms or pdfs of P, PET, and ET. It may be also possible to identify years with ΔS>0, for which ETwb > PET. I would also recommend plotting the time series of PET in Figure 8.

Section 4.1: It seems trivial to calibrate w with observed P, PET, and Q, and then compare to ETwb = P − Q. Is w calibrated for each catchment based on long-term mean annual P, PET, and Q? Do you have any arguments for the underestimation of ET estimates from the abcd model at multi-annual timescales (ETabcd)?

Section 4.3.2: As I understand it, ETwb − ΔS better represents actual ET than ETwb. Therefore, I think it would be more intuitive to compare ETwb − ΔS with ETBudyko, rather than ETwb with ETBudyko + ΔS.

Technical corrections

The text needs to be improved overall.

Define what you mean by humid and non-humid catchment at the introduction. References for studies about non-humid catchments are not really accurate; data from these studies also include humid catchments. Missing reference to Greve et al. (2016) for studies considering $\Delta$S.

Caption of Figure 1: The aridity index should be < 1, instead of > 1.

Criteria

1. Does the paper address relevant scientific questions within the scope of HESS?

Ok.

2. Does the paper present novel concepts, ideas, tools, or data?

Yes, but it could benefit from a different focus of the study. It is interesting for quantifying the influence of water storage changes in ET estimation, more than its importance being a new finding.

3. Are substantial conclusions reached?

Ok.

4. Are the scientific methods and assumptions valid and clearly outlined?

The main issue I find is with Budyko's assumption of steady-state conditions.

5. Are the results sufficient to support the interpretations and conclusions?

Ok.

6. Is the description of experiments and calculations sufficiently complete and precise to allow their reproduction by fellow scientists (traceability of results)?

Data is not accessible. I would strongly encourage to report all relevant data, e.g. long-term average P, PET, Q, catchment coordinates, etc, in supplementary material. The

methods could also benefit from a more detailed explanation.

7. Do the authors give proper credit to related work and clearly indicate their own new/original contribution?

Yes. Some additional references should be included, e.g. missing reference to Greve et al. (2016) for studies considering $\Delta$S. Include Zhang et al. (2004), since Fu (1981) is in Chinese.

8. Does the title clearly reflect the contents of the paper?

Yes, but it should be written properly, e.g. "The effect of changes in water storage in humid catchments on evapotranspiration estimates from Budyko's framework and water-balance models".

9. Does the abstract provide a concise and complete summary?

Improve text. It would benefit from a more objective/quantitative way of presenting the results. Statements like "works fine" and "much improvement" are not too informative.

10. Is the overall presentation well structured and clear?

The structure is fine. It may help storyline to have a stronger focus on the annual over multi-annual timescale since the beginning of the paper.

11. Is the language fluent and precise?

No. It definitely needs to be improved. At the current state it is difficult to assess the scientific merit of the manuscript.

12. Are mathematical formulae, symbols, abbreviations, and units correctly defined and used?

Ok.

13. Should any parts of the paper (text, formulae, figures, tables) be clarified, reduced, combined, or eliminated?

It would be useful to have a more rigorous test for the statistical significance of the improvement with size shown in Figure 9. I would recommend to exclude figure 10b and the corresponding methods section.

14. Are the number and quality of references appropriate?

Could be improved. See comments from other reviewers.

15. Is the amount and quality of supplementary material appropriate?

No. See for example point 6.

References

Gentine, P., P. D'Odorico, B. R. Lintner, G. Sivandran, and G. Salvucci, 2012: Interdependence of climate, soil, and vegetation as constrained by the Budyko curve. Geophys. Res. Lett., 39, 2–7, doi:10.1029/2012GL053492. Greve, P., L. Gudmundsson, B. Orlowsky, and S. I. Seneviratne, 2016: A two-parameter Budyko function to represent conditions under which evapotranspiration exceeds precipitation. Hydrol. Earth Syst. Sci., 20, 2195–2205, doi:10.5194/hess-20-2195-2016. Roderick, M. L., and G. D. Farquhar, 2011: A simple framework for relating variations in runoff to variations in climatic conditions and catchment properties. Water Resour. Res., 47, 1–11, doi:10.1029/2010WR009826. Zhang, L., K. Hickel, W. R. Dawes, F. H. S. Chiew, A. W. Western, and P. R. Briggs, 2004: A rational function approach for estimating mean annual evapotranspiration. Water Resour. Res., 40, 1–14, doi:10.1029/2003WR002710.

---

## Author Comment (AC1) · 9 Jun 2017

The writing of this paper needs a lot of improvement, and can benefit strongly from a native speaker rigorously rewriting it. I made a long list of suggestions for the first 50 sentences, but this list is not exhaustive (even not for this small part of the manuscript). In its current stage I cannot fully judge the scientific merit of the contribution because the I do not always understand the interpretation of generated results. The paper does not address anything sufficiently novel. The main conclusion that water storage changes can be significant at annual time-scales is already well known. In addition, some of the methods that the paper uses to estimate ET (Budyko and a water balance

accounting) have been extensively reported to be only applicable at longer time-scales. Consequently, I do not see why the paper is relevant to HESS. The manuscript is not based on sound reasoning that can be easily followed. While this may be a linguistic issue, since the authors are (I assume) no native English speakers, it hinders me from properly judging the scientific merit of this work.

Thank you. 1. For the content, we have rewritten and re-organized our manuscript.

a) We have added the result about the effect of $\Delta S$ on annual ET estimation based on Budyko framework in section 3.3. It shows that almost no improvement has been made in annual ET estimation based on the extended Budyko equation (Figure 6), which uses P-$\Delta S$ as 'equivalent' precipitation at monthly timescale through high R2 achieved (Figures 5 and 6), which is due to the seasonal pattern within the year.

<Figure 5 here> <Figure 6 here>

b) The effect of $\Delta S$ on annual ET validation based on water balance equation is shown in section 3.4. This section shows that much improvement has been made when comparing ETbudyko + $\Delta S$ with ETwb. And ignoring the variation of annual $\Delta S$ increases the variability of real ET and leads to large deviation in modelled ET assessment in humid region. This provides an acceptable explanation for the poorly estimated annual ET and reveals the important role of annual $\Delta S$ in ET estimation and validation in humid catchments.

c) We get the conclusion that the common practice of ignoring annual $\Delta S$ in water balance, can lead to larger deviation in estimated ET assessment in humid catchments. Without reliable $\Delta S$, ET estimation in humid catchments remains a challenge in bridging our gap in our knowledge of the hydrologic cycle.

2. The conclusion from the previous version, we may have put more words on the conclusion that water storage changes can be significant at annual time-scales, which is already well known, as we have summarized in the introduction in the revised version.

We meant to illustrate the effect of using ETwb to validate the modelled ET, large deviation would be achieved with small R2 between ETbudyko and ETwb. And attempts of improving modelled ET to meet annual ETwb would be inaccurate. Without reliable △S, ET estimation in humid catchments remains an important scientific challenge.

3. We all aware that the Budyko hypothesis can be used in steady state. But commonly, we use the Budyko equation at annual timescale, especially in arid and semiarid region. The estimated ET are validated against ETwb (△S~0) and satisfactory results can be achieved. In this study, we present the fact that, ignoring annual △S can affect ETwb more in humid catchments than that in non-humid region, and the same approach cannot be used in humid region at annual timescale. And more importantly, attempts of improving modelled ET to meet annual ETwb would be inaccurate.

4. The language. We feel terribly sorry for all the inconvenience we made here. We have sought help from a native speaker, who is a postdoctor in hydrology, to revise the manuscript. Much improvement has been made in the revised version. Hoping that this version is readable and interesting to you.

The paper does not review previous work appropriately. Statements like "While in humid region, the quantity of research are limited (Tekleab et al., 2011;Zhang et al., 2012; Carmona et al., 2016)" do not represent the vast work that has been done in ET research. This also related to the previously addressed issue that the novelty of this paper is not sufficient.

Thank you for your invaluable suggestions! We have revised the manuscript accordingly. We mean that annual ET estimation in humid catchments is relatively limited on areas containing humid catchments only when compared with that in arid and semi-arid catchments. Vast intra-annual ET estimation has been done in areas containing both humid and non-humid catchments. I didn't make this clear due to language issue. Sorry again. The revised version is in lines 64~98.

Detailed comments: Title: Avoid abbreviations in the title, and write evapotranspiration

instead of ET Title: the title should be include the time-scale(s) that the paper reports on (e.g. annual) Title: "for evapotranspiration estimations" instead of "in ET estimation" Title: "water balance models" is unclear. I suggest to use more commonly used wording. Title: "based on the Budyko framework" instead of "based on Budyko framework"

Done, we have revised our title as "The effect of water storage change on annual evapotranspiration estimation in humid catchments based on the Budyko framework". Thank you.

Line 26: "water-energy budget research" is unclear. Line 26: "resources" instead of "resource" Line 26-27: "while it", be explicit where "it" refers to. Line 26 – 28: this sentence needs to be reformulated. Line 28: include the timescale of ET estimates. Line 29: "located in" instead of "over" Line 28 – 32: "Here we : : : time scale". This sentence should be completely rewritten. Consider to break down this information into two separate (shorter) sentences. Line 32: "and we" is not a way to start a sentence. Line 34: "works fine" is vague Line 34: replace "in" by "for" Line 35: be more specific than "well" Line 36: "of this poor" instead of "for this poorly" Line 37: "the neglecting of dS" should be replaced by something like "the assumption that storage changes are negligible" Line 37: "real" is redundant Line 37: "increases" instead of "enlarges" Line 37: specify if "it" refers to "neglecting storage changes" or "storage changes" Line 38: "Much improvement has been made" Line 47: replace "returns" for "equals" Line 49: "is captured by" is vague. Do you mean mean "controlled by" Line 50: replace "i.e.," by " as described by" Line 46: "over terrestrial surface" is redundant since terrestrial and land are repeated later in the sentence. Line 50: Why is there no citation to the original Budyko paper/book? Line 52: the limitation you refer to are explained in the original Budyko equation, there is no need to refer to other studies here as long as this original publication is cited. Line 51: ", and it" it is unclear what "it" refers to + you should start this as a new separate sentence.

Thank you. We have revised as suggested. Since the whole manuscript has been rewritten, and some of the changes are untraceable in the revised manuscript.

Line 37-39: This sentence should be rigorously rewritten: "Much improvement has been made when comparing estimated ET + $\Delta$S with those ETwb, and the bigger the catchment area is, the better this improvement can be." The message is clear, but it the sentence is not English.

Thank you for this comment. This sentence has been removed from the abstract and the results, since it is the major contribution of this article.

Line 39-40: what does "an acceptable explanation" mean? Because of the confused English I am unsure how "certain" this attribution is.

Thank you. We have removed this description and replaced with "This provides a possible explanation for the poorly estimated annual ET and reveals the important role of annual $\Delta$S in ET estimation and validation in humid catchments.", in lines 39-41.

Line 41-43: "We highlight : : : humid catchments." Or "Overall this highlights that storage changes affect evapotranspiration at the annual time-scale and should therefore be accounted for."

Thank you for this comment. We have rewritten our highlights, "We highlight that the common practice of ignoring annual $\Delta$S in water balance, can lead to larger deviation in estimated ET assessment. Without reliable $\Delta$S, ET estimation in humid catchments remains a challenge in bridging our gap in our knowledge of the hydrologic cycle.", to provided more clear highlight more readers.

Please also note the supplement to this comment:
http://www.hydrol-earth-syst-sci-discuss.net/hess-2017-151/hess-2017-151-AC1-supplement.zip
* * *
[Figure]

[Figure]

R² plot showing box plots across months (Jan–Dec).

Legend:
- ◇ 25%~75%
- ⊥ 10%~90%
- — Median
- × Max/Min
- □ Mean

**Fig. 1.** Figure 5 The box plot of R2 between monthly ETwb and ETbudyko using the extended Budyko equation, i.e., P-ΔS as equivalent P, and ΔS is obtained from abcd model.

[Figure]

**Fig. 2.** Figure 6 The R2 between ETwb and ETBudyko at monthly timescale and that aggregated to annual timescale in (a), and (b) the boxplot of R2 of this aggregated annual ETbudyko and the original R2 of annua

---

## Author Comment (AC2) · 9 Jun 2017

Response:

The authors appreciate the reviewer for helpful and constructive comments that improved our original manuscript. We have addressed the comments below and made corrections. The changes being made are marked in revised version in the manuscript.

The authors perform a multi-method assessment of ET in wet basins in Southern China and attempt to assess the role of assuming change in water storage as negligible (DS=0) in their calculations at the annual and muti-annual time scales. They also

propose that this assumption may be the cause of poor annual ET estimates from three models (Fu, abcd and Xin'anjiang model) when compared to the ET from the water budget estimate at the annual scale. They find that including the estimates of Delta S at the annual scale from the abcd model reduces the variability of ET predictions. I think the subject is interesting and is nowadays gaining a lot of attention due to the wide use of the Budyko-type studies. The use of the abcd model to calculate DS sounds like a good idea, and to compare ET Budyko with ET Budyko + $\Delta$S. The scientific basins of this study is sound, at least from what I managed to extract from the extremely-difficult-to-understand manuscript. The exploration of change in water storage is interesting and resourceful (Fig. 7-10). But I have some concerns that must be fixed in this manuscript:

1. Language. As it is now, I think the manuscript is unpublishable in HESS or any other decent scientific journal. The use of English is of poor quality. Some parts of the manuscript cannot be judged with scientific criteria because it is just impossible to understand what the authors are referring to. I myself do not have English as a native language, so I understand how frustrating it is to express your findings in another language. However, this manuscript needs to be written from beginning to end with the aid of a native English speaker. I recommend taking it to a professional writer or similar. I started fixing grammatical issues, but then I realized there was no case in doing this and I should better focus on the science.

It is really nice of you for being so considerate but it is our mistake. We feel terribly sorry for all the inconvenience we made here. We have sought help from a native speaker, who is a postdoctor in hydrology, to revise the manuscript. Much improvement has been made in the revised version in hoping that this version is readable and interesting to you.

It is worth mentioning that we have added the result about the effect of $\Delta$S on annual ET estimation based on Budyko framework in section 3.3. It shows that almost no improvement has been made in annual ET estimation based on the extended Budyko

[Figure]

equation (Figure 6), which uses P-ΔS as 'equivalent' precipitation at monthly timescale through high R2 achieved (Figures 5 and 6), which is due to the seasonal pattern within the year. This further supports our conclusion that the common practice of ignoring annual ΔS in water balance, can lead to larger deviation in estimated ET assessment in humid catchments. Without reliable ΔS, ET estimation in humid catchments remains an important scientific challenge.

<Figure 5 here> <Figure 6 here>

2. Literature review on water storage and Budyko needs to be improved and updated. The authors have omitted important pieces of research dealing with the importance of water storage changes within the Budyko framework and specifically in "wet" regions. Some examples, -[Moussa and Lhomme, 2016] – This study should give some insight on the possible mathematical formulations that the authors could further explore, apart from their multimethod assessment. [Jaramillo and Destouni, 2015] – I know that the basins that the authors are studying are "humid", so irrigation is highly unlikely. However, flow regulation by water impoundment in reservoirs or water transfers affect the evaporative ratio ET/P in the long term, and it is not due to the water stored in the reservoirs as the authors suggest by citing Mao (2016). They find that flow regulation acts like a proxy of land and water use that can explain ET/P changes more than PET/P changes. [Destouni et al., 2013] – Budyko analysis cannot get "wetter" than in cold and wet Sweden. This study shows that accounting for surface water storage changes at the annual scale does not change long-term trends in basin-scale ET/P. [Gudmundsson et al., 2016] – They find that changes in water availability are only dominated by changes in the aridity index in very humid climates

We are really appreciate your helpful and constructive comments. We have rewritten the introduction as suggested, and detailed review about water storage change is in lines 68-80 and liens 99∼116 in the revised introduction. The recommended references are really helpful and we have cited some of them, along with some new ones in the text.

3. I think the authors need to be clearer in what they refer with DeltaS. Moisture S and storage S. How can we know which is which. Is ïЁŻADЁĞ $\Delta S = \Delta S + \Delta G$.

Thank you for your suggestion. The $\Delta S$ is the soil moisture change plus ground water change. We have revised this as suggested in lines 72-73, and detailed introduction for China in lines 364-371 in revised manuscript. As for soil moisture change and ground water change, we use symbols Smt and Gt in the abcd model in the revised manuscript to show the difference. Thank you.

4. I think they should justify their work in a better way. Saying that the effect of storage change in ET calculations in wet regions is scarce is just not true.

Excellent advice to the point. Thank you. We have altered this (lines 64-80) and some other sentences that are vague and confusion, e.g., the highlight. "We highlight that the common practice of ignoring annual $\Delta S$ in water balance, can lead to larger deviation in estimated ET assessment. Without reliable $\Delta S$, ET estimation in humid catchments remains a challenge in bridging our gap in our knowledge of the hydrologic cycle."

We originally wanted to express that "there is currently limited research containing humid catchments only, and most inter- and intra-annual ET estimation research are focusing on areas containing both humid and non-humid catchments." Sorry about our mistakes as such.

5. What is the multiannual scale? How did you measure DS (storage, not moisture) at this scale? How is it different from the annual scale? This is not clear at all. You never specify the periods

The multiannual timescale is the multi-year averages, here is the averages of P, Q and ET for ~50 years. The $\Delta S$ is the soil moisture change plus ground water change. At multiannual timescale, $\Delta S = P - Q_{obs} - ET_{Budyko}$, and the $\Delta S$ is quite small (averages around 1mm). While at annual timescale, we obtained the $\Delta S$ from abcd output, and variation of this annual $\Delta S$ is relatively large, -50 ~ +50 mm/yr, and the values in some

years are larger than this range. Thank you.

6. Line 420 to 421. The improvement of R2 from 0.02 to 0.58 sounds quite extraordinary. You should show this scatter plot due to its importance

Done! We have added it in Figure 9b as suggested. Thank you.

<Figure 9 here>

Other issues: 49. controlled instead of captured 58. around the globe: : : and scales like 60. region(s), research is limited 71 Delete "exist" 81-82 This is obvious. 86 theoretical ET 89 specifically 105. Again, this is because not all drivers of change are taken into account with Budyko-type models. 111 "It would prone"? 183 proved 257. Excellent? Change this word 282 Terrestrial feature? 306. You should state p-values for all R2 values that you give 356 "launched"

Done. We have revised as suggested. Since the whole manuscript has been rewritten, and some of the changes are untraceable in the revised manuscript. But we have learned quite a lot during the process. Thank you.

303. Why do you use the NSE. For what? Explain

We use NSE to evaluate the runoff simulation, and it is just my favor in the first. We then take a deeper thought, and it is unnecessary since there are similarity between NSE and R2. So we use R2 instead throughout the text. Thank you.

90. The Mao study has one big problem. They calculate $\Delta"ET"= \Delta P-\Delta R-\Delta S$ and not $\Delta"ET"=\Delta P-\Delta R-\Delta(\Delta S)$, which is incorrect. That is why they get such a big effect of storage change in the ET calculations.

Sounds reasonable and refreshing, and it is definitely worth a deeper thought. But the conclusion like "the $\Delta S$ is not approximate zero annually" can be concluded, but we have removed the detailed description from the text. Thank you.

64-66 This is because these models do not account for all drivers of change in

ET/P. For instance, look at [Jaramillo and Destouni, 2014]. 74% of all movements in Budyko space cannot be explained by only changes in PET/P during long periods of time. Jaramillo, F., and G. Destouni (2014), Developing water change spectra and distinguishing change drivers worldwide, Geophys. Res. Lett., 41(23), 8377–8386, doi:10.1002/ 2014GL061848.

In our opinion, the poorly estimated annual ET is due to the ignoring of the variation of annual $\Delta S$ in ETwb in humid region, as we have presented the analysis in sections 4.3 and 4.4. Besides, at multi-annual timescale, the Budyko equation can well estimate ET as $\Delta S$ is approximately zero, which can be well explained by PET/P. Besides, the vast research in arid and semiarid region have shown that, the Budyko equation can well estimate ET at annual and multi-annual timescales when validated against ETwb since their $\Delta S$ can be seen as zero in ETwb.

Figure 6- Something does not fit here. ET+S+Q=ET/P?? Sometimes I feel that there is a confusion between variability, variance and R2. Please check this along the manuscript.

ET+$\Delta S$ +Q=P. As below, the Figure 6 is now Figure 7 in revised manuscript. The ETwb is P-Qobs when we ignored the variation of annual $\Delta S$, as common practice. Then the ETbudyko is validated against this ETwb, and the larger this annual $\Delta S$ is, the greater the bias will be to the assessment. We stop using variance in the context to avoid such confusion. Thank you for your suggestions.

<Figure 7 here>

53. What is humid= PET/P<1?

The humid catchments we used are defined by aridity index (PET/P) <1, line 215.

61-64 Could not understand this

We have deleted this sentence since it is deviated from our major purpose. In the submitting version, we want to show that, the proportional relationship exists in humid

catchments, but the estimated ET is not well when compared with ETwb.

224 Is DS = DG + DS? You have to differentiate storage from soil moisture, they are both S and confuses.

We have revised this and use symbol Sm to represent soil moisture change. Thank you truly.

342-349 Improve language. Impossible to understand due to the language.

Thank you and we have rewritten these sentences, lines 360∼363. "From another perspective, the neglecting annual $\Delta S$ in water balance has prone to errors associated with ungauged subsurface runoff transfer in humid catchments. Therefore it produces relatively unreliable ETwb as real ET in hydrology and the assessment of modelled ET. "

Caption of Figure 1: The aridity index should be < 1, instead of > 1.

Done, sorry about this mistake. Thank you.

Please also note the supplement to this comment:
http://www.hydrol-earth-syst-sci-discuss.net/hess-2017-151/hess-2017-151-AC2-supplement.zip
* * *
[Figure]

The box plot figure shows R² values with a legend indicating 25%~75%, 10%~90%, Median, Max/Min, and Mean.

**Fig. 1.** Figure 5 The box plot of R2 between monthly ETwb and ETbudyko using the extended Budyko equation, i.e., P-$\Delta$S as equivalent P, and $\Delta$S is obtained from abcd model.

[Figure]

**Fig. 2.** Figure 6 The R2 between ETwb and ETBudyko at monthly timescale and that aggre-gated to annual timescale in (a), and (b) the boxplot of R2 of this aggregated annual ETbudyko and the original R2 of annua

[Figure]

**Fig. 3.** Figure 9 The annual time series of PET, ETwb, ETbudyko and ETBudyko + ΔS over 1957-2013 for the selected typical catchment (a), and (b) the comparison between ETbudyko, ETBudyko + ΔS against ETwb in t

Water Limit

ET/P

Energy Limit

Q_obs

S

ET_budyko

P-Q_obs

PET(t)/P(t)

PET/P

0

**Fig. 4.** Figure 7 The schematic of $\triangle S$ in Budyko equation in humid catchments (energy limited). ETbudyko is estimated based on given P and PET, and validated against ETwb, i.e., P-Qobs-$\triangle S$ where $\triangle S \sim 0$.

---

## Author Comment (AC3) · 9 Jun 2017

Response:

The language should be largely improved. There are many typos and grammatical errors throughout the text. But even more problematically, the wording is often awkward and difficult to understand, and I actually couldn't understand the exact meaning of several sentences. Some reviewers made valuable suggestions to improve this. However, I think the authors should also consider seeking the advice of a native or fluent English speaker, or possibly using some professional language services.

[Figure]

We feel terribly sorry for all the inconvenience we made here. We have sought help from a native speaker, who is a postdoctor in hydrology, to revise the manuscript. Much improvement has been made in the revised version. Hoping that this version is readable and interesting to you.

General comments

This study analyzes the effect of changes in water storage at annual and multi-annual timescales for humid catchments in China. The authors perform a quantitative comparison of evapotranspiration (ET) estimates from the water balance, Budyko's framework, and two other models. They show that it is erroneous to obtain annual ET estimates from the water balance when neglecting changes in storage. I find the approach interesting, but numerous points need to be addressed. Particularly, the quality of the text needs to be greatly improved before being able to thoroughly assess the scientific merit of the study.

Thank you for your invaluable comments/suggestions sincerely! Detailed point to point response is as below. It is worth mentioning that we have added the result about the effect of $\Delta$S on annual ET estimation based on Budyko framework in section 3.3. It shows that almost no improvement has been made in annual ET estimation based on the extended Budyko equation (Figure 6), which uses P-$\Delta$S as 'equivalent' precipitation at monthly timescale through high R2 achieved (Figures 5 and 6), which is due to the seasonal pattern within the year. This further supports our conclusion that the common practice of ignoring annual $\Delta$S in water balance, can lead to larger deviation in estimated ET assessment in humid catchments. Without reliable $\Delta$S, ET estimation in humid catchments remains an important scientific challenge.

<Figure 5> <Figure 6>

Specific comments

It is not right to assume the validity of Budyko for annual timescales. The sentence:

"Subsequently analysis on annual water-energy balance have proofed that the Fu's equation can be used in both long-term and annual water-energy balances in nonhumid catchments (Yang et al., 2007) and humid catchments as well (Tekleab et al., 2011; Xu et al., 2013)", is not accurate. Both Tekleab et al. (2011) and Xu et al. (2013), together with many other studies (e.g. Gentine et al. 2012; Roderick and Farquhar 2011) indicate that the validity of Budyko's framework requires steady-state conditions, which are generally achieved by using data at time scales significantly longer than 1 year.

We agree. The Budyko hypothesis can be well applied at steady state. But the problem here is that, vast research have shown that satisfactory results have achieved in annual ET estimation in arid and semiarid region when using Budyko equation and then validating against ETwb ($\Delta S \sim 0$). So attempts have been made to improve the modelling so as to meet this ETwb in humid region as well, which is not right based on our conclusion," ignoring the variation of annual $\Delta S$ increases the variability of real ET and leads to large deviation in modelled ET assessment in humid region." As for the introduction, we have rewritten this since many sentences are very confusing.

I think it would be better if the storyline focuses from the beginning on the issue of neglecting $\Delta S$ for ETwb.

Excellent advice to the point. Thank you. We have revised the introduction and results in our manuscript as suggest. We are inspired to present the result of the effect of $\Delta S$ on annual ET estimation based on the Budyko framework in section 4.3, to further prove that ignoring the variation of annual $\Delta S$ in ETwb leads to large deviation in modelled ET assessment in humid region.

The approach for analyzing the inter-annual variability of ET needs to be clearer (equations 8 and 9). Is the effect of $\Delta S$ not accounted for? I believe it would be better to not include this section in the paper, and consequently Figure 10b. The authors already convey your point about the higher variability of ETwb compared to the other estimates

of ET in Figure 10a. They could strengthen this argument by showing the histograms or pdfs of P, PET, and ET. It may be also possible to identify years with $\Delta S > 0$, for which ETwb > PET. I would also recommend plotting the time series of PET in Figure 8.

Indeed. We have removed this part from the manuscript since they are less relevant to the results. We have added the time series of PET in Figure 9a.

Large differences are among the ranges of variabilities of P, PET, ETbudyko, and ETwb, neither the plot of their pdf nor the plot of cdf looks fine, we use Figure 11 instead. The cdf of PET, ETbudyko, and ETwb is as below,

<Figure 9 here> <Figure 11 here>

Section 4.1: It seems trivial to calibrate w with observed P, PET, and Q, and then compare to ETwb = P − Q. Is w calibrated for each catchment based on long-term mean annual P, PET, and Q? Do you have any arguments for the underestimation of ET estimates from the abcd model at multi-annual timescales (ETabcd)?

Thank you for your comments. The w is calibrated using annual P, PET and Qobs for each catchment, and we have lessen the related description, lines 251-254. The underestimation of multi-annual ETabcd, we think it is mainly due to the bias in calibrated runoff. On the other hand, the Budyko equation is the major approach we used here, and multi-annual ETabcd estimation is redundant , so we remove it from section 4.1.

Section 4.3.2: As I understand it, ETwb − $\Delta S$ better represents actual ET than ETwb. Therefore, I think it would be more intuitive to compare ETwb − $\Delta S$ with ETBudyko, rather than ETwb with ETBudyko + $\Delta S$.

We have thought about it. But we insist that using ETwb would be better than ETwb − $\Delta S$. Because P and Qobs are observed and reliable while $\Delta S$ is the model output, using ETwb − $\Delta S$ would be less accurate. On the other hand, when comparing ETwb with ETBudyko + $\Delta S$, the ETwb is purely observed and reliable.

The text needs to be improved overall.

We have sought help from a native speaker. If further improvement is needed, we'll go to the professional language services for help.

Define what you mean by humid and non-humid catchment at the introduction. References for studies about non-humid catchments are not really accurate; data from these studies also include humid catchments. Missing reference to Greve et al. (2016) for studies considering ∆S.

We take catchments with aridity index (PET/P) <1 as humid catchment, and we have defined it in the introduction. We use arid and semiarid in most cases instead of non-humid to avoid such confusion. Some of cited reference are indeed containing both humid and arid, semiarid catchments, and we have revised this in our manuscript, e.g., lines 81∼84. And the missing reference has been added in the introduction, sorry about this.

Caption of Figure 1: The aridity index should be < 1, instead of > 1.

Done, sorry about this mistake. Thank you.

Please also note the supplement to this comment:
http://www.hydrol-earth-syst-sci-discuss.net/hess-2017-151/hess-2017-151-AC3-supplement.zip

———————————————————

[Figure]

Fig. 1. Figure 5 The box plot of R2 between monthly ETwb and ETbudyko using the extended Budyko equation, i.e., P-ΔS as equivalent P, and ΔS is obtained from abcd model.

[Figure]

**Fig. 2.** Figure 6 The R2 between ETwb and ETBudyko at monthly timescale and that aggregated to annual timescale in (a), and (b) the boxplot of R2 of this aggregated annual ETbudyko and the original R2 of annua

[Figure]

**Fig. 3.** Figure 9 The annual time series of PET, ETwb, ETbudyko and ETBudyko + ΔS over 1957-2013 for the selected typical catchment (a), and (b) the comparison between ETbudyko, ETBudyko + ΔS against ETwb in t

[Figure]

**Fig. 4.** Figure 11 The spatial distribution of variabilities of ETBudyko, ETwb and PET in humid catchments over China in (a), (b) and (c), respectively, and their statistics information accompanied by the vari